# Functional Characterization of a Phf8 Processed Pseudogene in the Mouse Genome

**DOI:** 10.3390/genes14010172

**Published:** 2023-01-08

**Authors:** Joannie St-Germain, Muhammad Riaz Khan, Viktoriia Bavykina, Rebecka Desmarais, Micah Scott, Guylain Boissonneault, Marie A. Brunet, Benoit Laurent

**Affiliations:** 1Research Center on Aging, Centre Intégré Universitaire de Santé et Services Sociaux de l’Estrie-Centre Hospitalier Universitaire de Sherbrooke, Sherbrooke, QC J1H 4C4, Canada; 2Department of Biochemistry and Functional Genomics, Faculty of Medicine and Health Sciences, Université de Sherbrooke, Sherbrooke, QC J1H 5N4, Canada; 3Department of Pediatrics, Medical Genetics Service, Université de Sherbrooke, Sherbrooke, QC J1H 5N4, Canada; 4Centre de Recherche du Centre Hospitalier Universitaire de Sherbrooke (CRCHUS), Sherbrooke, QC J1H 5N4, Canada

**Keywords:** pseudogene, PHF8, histone demethylase, mouse genome, testis

## Abstract

Most pseudogenes are generated when an RNA transcript is reverse-transcribed and integrated into the genome at a new location. Pseudogenes are often considered as an imperfect and silent copy of a functional gene because of the accumulation of numerous mutations in their sequence. Here we report the presence of *Pfh8-ps*, a *Phf8* retrotransposed pseudogene in the mouse genome, which has no disruptions in its coding sequence. We show that this pseudogene is mainly transcribed in testis and can produce a PHF8-PS protein in vivo. As the PHF8-PS protein has a well-conserved JmjC domain, we characterized its enzymatic activity and show that PHF8-PS does not have the intrinsic capability to demethylate H3K9me2 in vitro compared to the parental PHF8 protein. Surprisingly, PHF8-PS does not localize in the nucleus like PHF8, but rather is mostly located at the cytoplasm. Finally, our proteomic analysis of PHF8-PS-associated proteins revealed that PHF8-PS interacts not only with mitochondrial proteins, but also with prefoldin subunits (PFDN proteins) that deliver unfolded proteins to the cytosolic chaperonin complex implicated in the folding of cytosolic proteins. Together, our findings highlighted PHF8-PS as a new pseudogene-derived protein with distinct molecular functions from PHF8.

## 1. Introduction

First described in 1977 [1], pseudogenes are considered as a defective copy of a functional gene that has lost its ability to produce a functional RNA or protein. Pseudogenes contain numerous alterations in their sequences, such as lack of promoter, presence of premature stop codons, or frameshift mutations that lead them to be functionally inert. As paralogs of ancestral genes, pseudogenes can be classified as either unitary, polymorphic, duplicated (unprocessed), or retrotransposed (processed) pseudogenes depending on the mechanisms they have been generated through during genome evolution [2,3]. Unitary pseudogenes are created via spontaneous mutations within a gene that prevent either its transcription or translation [4]. Polymorphic pseudogenes are coding genes that became pseudogenic due to the presence of a premature stop codon in the reference genome [3]. A polymorphic pseudogene exists as an allele along with the functional gene at the same locus. Duplication of DNA segments, which is essential for the evolution of complex genome to generate gene families, can also produce duplicated genes that loose their original functions due to either nonsense mutations or the absence of promoters or enhancers [5]. Duplicated pseudogenes usually retain the intact exon-intron structure of their parental genes. Finally, other pseudogenes can be generated when an RNA transcript is reverse-transcribed and integrated into the genome at a new location. These retrotransposed or “processed” pseudogenes generally lack introns and contain a poly-A tail towards the 3′ end [6]. It is estimated that more than 70% of pseudogenes have a retrotransposition-linked origin (processed), and the rest arose by gene duplication or mutation [7]. Pseudogenes are present in a wide range of species (e.g., plants and bacteria) and are particularly abundant in mammals [8]. Indeed, with an estimated number of 15,000 in the human genome, pseudogenes are almost as numerous as protein-coding genes [7,9]. Most pseudogenes are transcriptionally silent, either due to mutations in their promoter or integration in repressed regions of the genome [10]. They are also considered as a neutral sequence because of the accumulation of numerous mutations in their sequence. However, there is growing evidence that some pseudogenes are transcribed and active like PTEN, GAPDH, and Oct4 pseudogenes [11,12,13]. The estimated proportion of transcribed pseudogenes varies from 2% to 20% in the human genome [14]. Pseudogenes have tissue-specific expression patterns [14], and their expression can vary under specific conditions such as tumorigenesis [15,16]. Since their discovery, pseudogenes have raised the question of the importance for mammals to retain genes that do not produce a functional RNA or protein. Once considered “junk DNA”, many pseudogenes have now been reported to play key roles in health and diseases [3,17,18]. Pseudogenes serve as microRNA decoys to sponge microRNAs targeting parent genes, and hence form a regulatory pair with their parental genes [16]. Pseudogenes can also sponge RNA-binding proteins that regulate RNA stability of their parental genes [19], or facilitate 3D chromatin interactions in an RNA-independent manner [20]. Finally, some pseudogenes have no open reading frame (ORF) disruptions in their sequence and can produce new proteins [21,22,23].

Gene expression is transcriptionally controlled by the presence of active or repressive post-translational modifications on histone tails that regulate chromatin structure and function [24]. For instance, methylation of lysine residues is deposited by histone methyltransferases (KMTs) and can be removed by histone demethylases (KDMs). There are two families of KDMs: the lysine-specific demethylase (LSD) family [25] and the Jumonji C (JmjC) domain-containing family [26]. PHD finger protein 8 (PHF8) is a histone demethylase containing a single N-terminal plant homeodomain (PHD) finger followed by the catalytic JmjC domain. Via its PHD finger, PHF8 can bind the tri-methylation of lysine 4 on histone 3 (H3K4me3), a histone mark typically enriched at the transcription start sites, and hence be recruited at its target promoters [27]. Via its JmJC domain, PHF8 can catalyze the demethylation of H3K9me1/2 [28,29] and H4K20me1 [30,31] and activate gene expression [32]. Through its dynamic modulation of histone methylation, PHF8 participates in cell cycle regulation [30]. Dysregulation of PHF8 expression has been reported in various types of cancers, suggesting a pro-oncogenic role as observed for many demethylases [33,34,35,36]. *PHF8* mutations that disrupt its enzymatic activity have been associated to X-linked intellectual disability (XLID) and craniofacial deformities such as a cleft lip/palate phenotype [31,37,38,39].

Here, we report the presence of a *Phf8* processed pseudogene in the mouse genome. We show that this intronless pseudogene is transcriptionally expressed in testis, and more specifically in spermatids, as well as that it encodes for a functional protein (PHF8-PS) that has 67% of homology with PHF8 protein. As PHF8-PS harbors a well-conserved JmjC domain, we functionally characterized its enzymatic activity and showed that PHF8-PS does not exhibit the intrinsic capability to demethylate H3K9me2 in vitro compared to PHF8. Surprisingly, PHF8-PS does not localize in the nucleus like PHF8 but rather is mostly located at the cytoplasm. Finally, our proteomic analysis of PHF8-PS- and PHF8-associated proteins revealed that PHF8 specifically interacts with proteins involved in meiotic chromosome condensation, while PHF8-PS is associated with not only mitochondrial proteins, but also with prefoldin subunits that deliver unfolded proteins to a cytosolic chaperonin complex implicated in the folding of cytosolic proteins. Together, our findings highlight PHF8-PS as a new pseudogene-derived protein with distinct molecular functions from the parental PHF8 protein.

## 2. Material and Methods

### 2.1. Plasmid Construction

The mouse *Phf8* transcript (NM_177201) was purchased from Origene (Cat#: MR223276) and subcloned into a pENTR/D-TOPO vector (Thermo Fisher Scientific, Waltham, MA, USA). The mouse pseudogene *Phf8* (*Phf8-ps*) (Ref Sequence:4921501E09Rik, NM_001009544) was cloned by PCR into a pENTR/D-TOPO vector (Thermo Fisher Scientific, Waltham, MA, USA), using mouse genomic DNA. Both *Phf8* and *Phf8-ps* sequences were then transferred by Gateway reaction into a pHAGE-HA/FLAG vector, using the Gateway LR Clonase II enzyme (Thermo Fisher Scientific, Waltham, MA, USA).

### 2.2. Cell Culture and Plasmid Transfections

The HEK293T and NIH3T3 cell lines were cultured in Dulbecco’s modified eagle’s medium (DMEM; with a glucose concentration of 25 mM) supplemented with 10% fetal bovine serum (FBS), 1% penicillin/streptomycin, and 1% sodium pyruvate at optimal culturing conditions 37 °C with 5% CO_2_. Cells were transfected using jetOPTIMUS^®^ DNA transfection reagent (Avantor, Radnor Township, PA, USA) according to the manufacturer’s protocol.

### 2.3. RNA Isolation and Quantitative PCR

Each frozen organ of C57BL/6J mice was crushed and powdered using the Bio Squeezer Snap Freeze Press system (BioSpec Products, Bartlesville, OK, USA). The subsequent powder was then fractionated and stored at −80 °C in multiple Eppendorf tubes. To isolate total RNA from the tissue powder, a hybrid technique combining the TRIzol reagent protocol (Invitrogen, Thermo Fisher Scientific, Waltham, MA, USA) and the Quick-RNA™ Miniprep Kit (Zymo Research, Thomas Scientific, Swedesboro, NJ, USA) was used. Powder was first homogenized in 1 mL of cold TRIzol and then 500 μL of chloroform:isoamyl alcohol 24:1 (Sigma Aldrich, St. Louis, MO, USA) was added. Samples were vortexed for 20 s and incubated 2 min at room temperature. The samples were then centrifuged at 10,000× *g* for 15 min at 4 °C. The aqueous phase, which contains RNAs, was transferred in a new Eppendorf tube. An equal volume of ethanol 100% was added, and the mixture was then loaded into a Zymo-Spin™ IIICG Column from the Quick-RNA™ Miniprep Kit. Further steps were conducted according to the manufacturer’s protocol. The RNA concentration was determined using a Nanodrop system (Thermo Fisher Sscientific, Waltham, MA, USA). Total RNA (1 μg) was used to generate cDNA using the Biorad iscript RT supermix RT-qPCR kit (Biorad, Hercules, CA, USA) following the manufacturer’s instructions. The expression of different target genes was validated by quantitative PCR (qPCR), using the Azur Cielo 3 system (Azure Biosystems, Dublin, CA, USA). The specific mouse primers used for the amplification are listed below:

*Gapdh* F: GGA GAA GGC CGG GGC CCA CTT G

*Gapdh* R: CAA TGC CAA AGT TGT CAT GGA TGA CC

*Phf8-ps* F: TGA AAG TGA GGC GGG AAA TGC TTG TGT A

*Phf8-ps* R: TGG AGT GTT GGG ATC TCA TCT GCC C

*Phf8* isoform 1 F: GCT CCA TGG AGT CCT AAA GCC CGT GTG

*Phf8* isoform 1 R: CTC TAA GTC ATC AGT AGC AGG GCC ACC

*Phf8* isoform 2 F: GGC ATG GCT CAG GCA AAT CGC AGC

*Phf8* isoform 2 R: CGA GTC TCT GCT TTG CTG TGG CCA GG

The reactions were performed with the PerfeCta SYBR Green SuperMix (QuantaBio, Beverly, MA, USA) as recommended by the manufacturer. Real-time PCR was performed with a hot start step 95 °C for 2 min followed by 40 cycles of 95 °C for 10 s, 60 °C for 10 s, and 72 °C for 20 s. Analysis was performed using Azur Cielo software (v1.0.0.285, Azure Biosystems, Dublin, CA, USA). The relative expression of genes was normalized by that of *Gapdh*.

### 2.4. Cell Fractionation and Protein Quantification

Cells were washed with PBS 1x, resuspended in a fractionation buffer (20 mM HEPES pH 7.4, 10 mM KCl, 2 mM MgCl_2_, 1mM EDTA, 1 mM EGTA, 1 mM DTT, and 1× protease inhibitor cocktail mix) and incubated on ice for 15 min. Cell suspensions were then passed through a 27 G needle at least 10 times and left on ice for another 20 min. After incubation, cell suspensions were centrifuged at 3000 rpm for 5 min, and the supernatant corresponding to the cytoplasmic fraction was collected in a new tube. The nuclear pellet was washed with 500 μL of fractionation buffer and then passed through a 25 G needle 10 times, followed by a centrifugation at 3000 rpm for 10 min at 4 °C. The supernatant was discarded, and the nuclear pellet was then resuspended in TBS with 0.1% SDS, sonicated for 12% for 3 s, and then boiled at 95 °C for 5 min. Absolute protein quantification of the cytoplasmic and nuclear fraction was performed by Bicinchoninic acid assay (BCA) using the Pierce™ BCA Protein Assay Kit (Thermo Fisher Sscientific, Waltham, MA, USA) according to the manufacturer’s parameters. The absorbance was read at 562 nm using a spectrophotometer, and the absolute quantification was calculated.

### 2.5. Immunoblotting

The proteins were resuspended in 1× Laemmli (62.5 mM Tris pH 6.8, 25% glycerol, 2% SDS, 0.1% bromophenol blue, 100 mM DDT) and then boiled for 5 min at 95 °C. Samples were then analyzed on a 10% SDS-PAGE gel in 1× running buffer at 100 V for around 1 h 30 min. Proteins were transferred on a nitrocellulose membrane (Whatman, Maidstone, UK) with 1× Tris-glycine buffer at 100 V for 1 h. The membranes were blocked with 5% m/v powdered milk dissolved in 1× PBS—0.1% tween for 30 min, and then washed three times for 5 min in PBS—0.1% tween with agitation at room temperature. Membranes were incubated overnight at 4 °C with the primary antibody (Table 1). The next day, the membranes were washed three times for 5 min in PBS—0.1% tween at room temperature and incubated for 1 h with the horseradish peroxidase (HRP)-conjugated secondary antibody at room temperature (Table 1). After this incubation, the membranes were washed again three times for 5 min with PBS—0.1% tween. Results were visualized on an iBright machine (Thermo Fisher Scientific, Waltham, MA, USA).

For demethylase assay immunoblots, after the first antibody revelation (H3K9me2), both the primary and secondary antibodies were removed from the membrane with a stripping solution (1% SDS, 62.5mM Tris HCl pH 6.8, 0.8% β-mercaptoethanol) for 30 min at 45 °C. The membranes were then washed 5 min in PBS—0.1% tween, washed and blocked again with 5% m/v powdered milk dissolved in 1× PBS—0.1% tween for 30 min, and finally incubated overnight at 4 °C with a new primary antibody (H3).

### 2.6. Immunofluorescence

In total, 20,000 to 30,000 cells were seeded onto coverslips placed in 24-well plates. The next day, cells were transfected as described above. Then, 24 h post-transfection, cells were washed three times with ice-cold PBS and fixed using 4% paraformaldehyde (PFA) for 20 min at room temperature. After fixation, cells were washed again with PBS, and then permeabilizated with 0.15% Triton-X100 in PBS for 5 min on ice, followed by two washes with PBS. Next, we used 10% goat serum as a blocking agent. We incubated coverslips with primary antibodies diluted in goat serum (Anti-HA, ref #11867423001, Roche (Basel, Switzerland); Anti-Tom20, ref #42406S, Cell Signaling (Danvers, MA, USA); Anti-P4HB, ref #A19239, Abclonal (Woburn, MA, USA),; Anti-Calnexin, Abclonal (Woburn, MA, USA), ref#A15631) in a humid chamber overnight at 4 °C. The next day, the cells were washed three times with PBS and incubated with Alexa Fluor anti-rabbit and anti-mouse secondary antibodies diluted in goat serum (respectively ref #A-21244 and #A-11017, Thermo Fisher Scientific, Waltham, MA, USA) for 1 h at room temperature. Cells were then washed with PBS and stained with a DAPI solution for 10 min at room temperature. In the end, the coverslips were mounted on the glass slides for microscopy. High-quality images were captured using confocal microscopy (Zeiss LSM 880; Ziess, Oberkochen, Germany).

### 2.7. Immunoprecipitation

For immunoprecipitation assays, HEK293 cells transiently overexpressing either HA/FLAG-PHF8 or HA/FLAG-PHF8-PS were suspended in a lysis buffer (0.5% NP-40, 300 mM NaCl, 20 mM HEPES, pH 7.4, 2 mM EDTA, and 1.5 mM MgCl_2_) and incubated for 30 min on ice before a centrifugation at 15,000 rpm for 20 min at 4 °C. Supernatants were collected and incubated with anti-FLAG M2 agarose beads (Sigma Aldrich, St. Louis, MO, USA) for 4 h at 4 °C. Following incubation, beads were washed with a lysis buffer with different concentrations of salt: three washes with a 300 mM NaCl lysis buffer to remove non-specific proteins, two washes with a 150 mM NaCl lysis buffer, and two washes with a 50 mM NaCl lysis buffer for equilibration. Immunoprecipitations were subsequently used for demethylase assays or for analysis by mass spectrometry (MS).

### 2.8. Demethylase Assays

Demethylase assays were carried out on 10 μg of calf histones in a 100 μL reaction volume with the following demethylase buffer: 50 mM Tris-HCl ph 7.5, 50 mM HEPES-KOH (pH 7.9), 50 mM NaCl, 1 mM MgCl_2_, 1 mM α-ketoglutarate, 2 mM ascorbic acid, and 100 μM ammonium iron (II) sulfate. Beads with immunoprecipitated PHF8 and PHF8-PS complexes were added to the demethylation reaction and incubated in a shaking thermocycler at 37 °C for 4 h. Reactions were finally boiled at 95 °C for 5 min, and histone proteins were further analyzed by immunoblotting.

### 2.9. Mass Spectrometry

*Sample preparation.* The beads were washed five times with 50 mM of ammonium bicarbonate and then resuspended in 100 μL of the same buffer. Proteins were digested by adding 1 μg of Pierce MS-grade trypsin (Thermo Fisher Scientific, Waltham, MA, USA) and incubated overnight at 37 °C with stirring (1250 rpm). The next day, digestion was stopped by adding formic acid (FA) to a final concentration of 1%. Beads were stirred at 1250 rpm for 5 min at room temperature, and then centrifuged at 2000× *g* for 3 min before the supernatant was harvested and transferred to a new low binding microtube. Beads were resuspended in 100 μL of buffer containing 60% acetonitrile (ACN) (Sigma-Aldrich, St. Louis, MO, USA) and 0.1% FA, and subsequently stirred at 1250 rpm for 5 min at room temperature. The supernatant was harvested carefully and combined with the previous one. Samples were thereafter concentrated by a centrifugal evaporator at 60 °C until complete drying (approximately 2 h) and resuspended in 30 μL of 0.1% trifluoroacetic acid (TFA) buffer (Sigma-Aldrich, St. Louis, MO, USA). Peptides were purified with ZipTip tips containing a C18 column (MilliporeSigma, Burlington, MA, USA) as recommended by the manufacturer. Eluted peptides were concentrated by centrifugal evaporator at 60 °C until complete drying (approximately 2 h) and then resuspended in 50 μL of 1% FA buffer. Peptides were next assayed using a NanoDrop spectrophotometer (Thermo Fisher Scientific, Waltham, MA, USA) and read at an absorbance of 205 nm. Peptides were then transferred to a glass vial (Thermo Fisher Scientific, Waltham, MA, USA) and stored at −20 °C until analysis by MS.

*LC-MS/MS analysis*. For LC-MS/MS, 250 ng of each sample were injected into an HPLC (nanoElute, Bruker Daltonics, Billerica, MA, USA) and loaded onto a trap column with a constant flow of 4 µL/min (Acclaim PepMap100 C18 column, 0.3 mm id × 5 mm, Dionex Corporation, Sunnyvale, CA, USA) and then eluted onto an analytical C18 Column (1.9 μm beads size, 75 µm × 25 cm, PepSep, Bruker Daltonics, Billerica, MA, USA). Peptides were eluted over a 2 h gradient of acetonitrile (5–37%) in 0.1% FA at 400 nL/min while being injected into a TimsTOF Pro ion mobility mass spectrometer equipped with a captive spray nano electrospray source (Bruker Daltonics, Bruker Daltonics, Billerica, MA, USA). Data were acquired using data-dependent auto-MS/MS with a 100–1700 m/z mass range, with PASEF enabled with a number of PASEF scans set at 10 (1.17 s duty cycle) and a dynamic exclusion of 0.4 min, m/z-dependent isolation window, and collision energy of 42.0 eV. The target intensity was set to 20,000 with an intensity threshold of 2500.

*Protein identification by MaxQuant analysis*. The raw files were analyzed using the MaxQuant version 2.0.3.0 software (Max Planck Institute of Biochemistry, Martinsried, Germany) and the Uniprot mouse proteome database (7 March 2021; 55,315 entries) with the addition of the sequence of PHF8-PS. The settings used for the MaxQuant analysis (with TIMS-DDA type in group-specific parameters) were: two miscleavages were allowed, the enzyme was Trypsin (K/R not before P), and variable modifications included in the analysis were methionine oxidation and protein N-terminal acetylation. A mass tolerance of 20 ppm was used for precursor ions, and a tolerance of 40 ppm was used for fragment ions. Identification values “PSM FDR”, “Protein FDR”, and “Site decoy fraction” were set to 0.05. Minimum peptide count was set to 1. Label-Free-Quantification (LFQ) was also selected with a LFQ minimal ratio count of 2. Both the “Second peptides” and “Match between runs” options were also allowed. Proteins positive for at least either one of the “Reverse”, “Only.identified.by.site”, or “Potential.contaminant” categories were eliminated, as well as proteins identified from a single peptide. Proteins were searched using a target-decoy approach against UniprotKB (Mus musculus, SwissProt, 2020-10) with the addition of the sequence of PHF8-PS. The false discovery rate (FDR) was set at 1% for peptide-spectrum match, as well as peptide and protein levels. Only proteins identified with at least two unique peptides were kept for downstream analyses. Protein interactions were scored using the SAINT algorithm. The fold change over the experimental controls and the SAINT probability scores were calculated as follows. The fold change was evaluated using the geometric mean of replicates and a stringent background estimation. The SAINT score was calculated using SAINTexpress, with experimental controls and default parameters. Proteins with a SAINT score of 1 and a fold change above 20 were considered highly confident interacting proteins. The network of PHF8- and PHF8-PS-interacting proteins was built using Python scripts (version 3.7.3) and the NetworkX package (version 2.4). The interactions from the STRING database were retrieved from their protein links’ downloadable file. Only interactions with a combined score above 750 were kept.

### 2.10. Peptide-Centric Proteome Analysis

Published MS datasets of testis (PXD011890 and PXD017284) were interrogated using a peptide-centric approach to identify spectra matching peptides unique to PHF8-PS. The PXD017284 dataset corresponds to the mouse germ cell proteome at nine stages of spermatogenesis [40], while the PXD011890 dataset corresponds to the phosphoproteome of mouse spermatids [41]. Spectra were searched using the PepQuery algorithm [42]. This peptide-centric approach compares annotation of each spectrum with peptides from mouse PHF8-PS to that with any protein from the reference protein database (*Mus musculus*, UniProtKB, 2021-01). Peptides from PHF8-PS were retrieved after in silico trypsin digestion, allowing for up to two miscleavages, a minimal peptide length of seven amino acids, and a mass from 400 to 6000 Da to mimic experimental conditions. Each peptide was queried using PepQuery, and we retrieved any peptide-spectrum match (PSM) that was better matched with our queried peptide from PHF8-PS than with any peptide with any protein from the reference protein database. For a PSM to be considered as confident, a *p*-value below 0.01 and a hyperscore higher than any peptide with or without any post-translational modification (PTM) from the reference protein database are needed.

## 3. Results

### 3.1. Phf8-ps, a Phf8 Processed Pseudogene in the Mouse Genome

The *Phf8* gene is located on chromosome X of the mouse genome (XqF4 region; 150,303,668–150,416,855 in GRCm39). The gene locus is 113.19 kb long and consists of 22 exons. According to the NCBI reference sequence database, *Phf8* gene produces several spliced variant transcripts. Phf8 transcript 1 (NM_177201) is 3732 bp long and contains 19 exons that encode a protein of 795 amino acids (88.9 kDa) (Figure 1A). Phf8 transcript 2 (NM_001113354) is 6359 bp long and contains 22 exons that encode a protein of 1023 amino acids (113.6 kDa) (Figure 1A). Compared to Phf8 transcript 1, this transcript contains an alternative spliced exon of 303 bp (exon 13) and a last exon (2519 bp) including a large 3′UTR of 2430 bp. Both isoform proteins encoded by these transcripts contain the PHD and JmjC domains since the inclusion of these alternative exons within this transcript does not alter the sequence integrity of these domains (Figure 1B). As reported in databases (e.g., GTEx Portal), other spliced variant transcripts can be generated from the *Phf8* gene; however, their levels of expression are very low compared to these two transcripts.

We also identified a *Phf8* pseudogene (*Phf8-ps*) located on chromosome 17 of the mouse genome (17qB1 region; 33,283,117–33,286,999 in GRCm39). This intronless pseudogene was predicted to produce a *Phf8-ps* transcript (NR_160433) that is 3883 bp long. Sequence alignment of *Phf8-ps* and *Phf8* transcripts indicated that the *Phf8* pseudogene might be derived from the retrotransposition of *Phf8* transcript 2 because of the presence of exon 13 within *Phf8-ps* (Figure 1A). Surprisingly, despite the accumulation of numerous mutations in *Phf8-ps* sequence, our sequence analysis revealed a potential ORF that encodes for a putative PHF8-PS protein of 908 amino acids (1021 kDa) (Figure 1A). The protein sequence alignment indicated that PHF8-PS protein had 67% of homology with the PHF8 protein encoded by transcript 2 (*Phf8* isoform 2) (Figure 1C). The PHD and JmjC domains of PHF8 are well-conserved in the PHF8-PS protein (75% and 87% of homology with PHF8, respectively) (Figure 1C), suggesting that PHF8-PS might also potentially act as a chromatin regulator by binding PHF8 target gene promoters and by demethylating histones.

### 3.2. Phf8-ps Is Transcribed and Encodes for a Protein in Testis

We next investigated whether *Phf8-ps* was an expressed or silent pseudogene. We collected different mouse organs, i.e., brain, heart, spleen, lung, liver, and testis. We specifically micro-dissected different areas of the brain since *PHF8* mutations were associated with XLID [31,37,38,39]. We first analyzed the expression of both Phf8 transcripts by quantitative PC. We showed that *Phf8* transcript 1 was predominantly expressed across all tissues compared to *Phf8* transcript 2, with the highest expression in the spleen, lung, and testis (Figure 2A). We next quantified the potential expression of Phf8-ps mRNA in the same tissues. Our results indicated that *Phf8-ps* was expressed at the RNA level, and that its expression was mainly detected in testis (Figure 2B). Given that *Phf8-ps* is an intronless pseudogene, we addressed the possibility of non-specific quantification of *Phf8-ps* due to residual genomic DNA contamination. We performed similar quantitative PCR on cDNA from testis samples prepared with or without reverse transcriptase and showed that *Phf8-ps* mRNA was not amplified in the samples without reverse transcriptase, confirming that *Phf8-ps* is indeed a transcribed pseudogene (Figure 2C). We next cloned the ORF of *Phf8-ps* into an expression vector with an HA/FLAG tag and transfected the obtained plasmid in the mouse NIH3T3 cell line to sequentially analyze the expression of *Phf8-ps* at protein level. As control, we performed the same experiment with the *Phf8* transcript 1, the most expressed *Phf8* splice variant in testis. We showed that cells transfected either with *Phf8* or *Phf8-ps* construct expressed a protein at the expected molecular weight (respectively 90 kDa vs 102 kDa), confirming that *Phf8-ps* can be translated into a protein in vitro (Figure 2D). To further support these results and confirm that *Phf8-ps* encoded a protein in vivo, we performed a peptide-centric proteome analysis on published mass spectrometry (MS) datasets to identify spectra matching peptides unique to PHF8-PS in testis. The first dataset (PXD017284) corresponded proteomic analyses of nine male germ cells populations encompassing differentiation steps ranging from spermatogonia to round spermatids [40], while the second one (PXD011890) consisted to the proteome of mouse spermatids [41]. Peptides from PHF8-PS were retrieved after in silico trypsin digestion, and each peptide was compared to peptides of any protein from the UniProtKB reference protein database to identify unique PHF8-PS peptides. Unique peptides for PHF8-PS were finally queried to our MS datasets. We identified a unique PHF8-PS peptide (SPSSVLGTVSDSPVSR) in the first MS dataset, more specifically in the haploid spermatid stage of spermatogenesis, confirming the presence of an endogenous PHF8-PS protein (Figure 2E). Together our results indicated that *Phf8-ps* was mainly transcribed in mouse testis and can encode for a protein both in vitro and in vivo.

### 3.3. PHF8-PS Does Not Have a Demethylase Activity

PHF8-PS has a conserved JmjC domain (87% of homology with PHF8) (Figure 1C) and hence could have a similar enzymatic activity by demethylating H3K9me1/2 or H4K20me1 like the parental PHF8 protein [28,29,30,31]. To address this hypothesis, we performed demethylation assays on bulk histones using HA/FLAG-tagged PHF8 and PHF8-PS proteins purified from HEK293 cells. We observed that PHF8 demethylated H3K9me2 as expected (Figure 3, left panel). The specificity of this demethylation was confirmed by carrying out the same experiment with a PHF8 catalytic mutant, PHF8 F279S. The mental retardation-linked F279S mutation impairs PHF8 histone demethylase activity [43]. As expected, we showed that PHF8 F279S was indeed catalytically inactive on the H3K9me2 substrate (Figure 3, left panel). We next determined whether PHF8-PS had an enzymatic activity on H3K9me2 similar to PHF8 and showed that PHF8-PS was unable to remove methyl groups of this modification (Figure 3, right panel). To ensure that this result did not depend on PHF8-PS stability, we performed similar demethylase assays with an increasing amount of PHF8-PS and confirmed that PHF8-PS did not have the intrinsic capability to demethylate H3K9me2 like the PHF8 parental protein (Figure 3, right panel).

### 3.4. PHF8-PS Is Mainly Located in the Cytoplasm

Since PHF8-PS did not have the intrinsic property to demethylate H3K9me2, we hypothesized that PHF8-PS could have a different subcellular localization and hence distinct functions in the cell. We first looked at nuclear localization signals (NLS) and nuclear export signals (NES) in the PHF8-PS protein using NLStradamus [44] and LocNES [45]. We identified a NES sequence in PHF8-PS (amino-acids 607-621), but we were unable to detect any NLS sequence (Figure 4A). Interestingly, computational tools detected three putative NLS sequences within the parental PHF8 protein, suggesting that PHF8-PS might have a different subcellular localization (Figure 4A). To support the computational predictions, we performed a cell fractionation assay on the mouse NIH3T3 cell line overexpressing either HA/FLAG-tagged PHF8 or PHF8-PS and analyzed the distribution of the proteins in the cytoplasmic and nuclear fractions by immunoblot. As expected, we observed that PHF8 was mainly detected in the nuclear fraction with only few proteins found in the cytoplasmic fraction (Figure 4B, middle panel) while PHF8-PS was exclusively found in the cytoplasmic fraction with no proteins in the nuclear fraction (Figure 4B, right panel). To further corroborate these results, we performed immunofluorescence on cells overexpressing either PHF8 or PHF8-PS to visualize the subcellular localization of each protein. We showed that PHF8 was strictly nuclear, while PHF8-PS was exclusively located in the cytoplasm (Figure 4C). Co-immunostaining with the mitochondrial marker Tom20 indicated that a small fraction of PHF8-PS proteins was colocalized with Tom20 at the mitochondria (Figure 4C). Similarly, co-immunostaining with the endoplasmic reticulum markers P4HB and calnexin showed that a fraction of PHF8-PS was also colocalized at the endoplasmic reticulum (Appendix A). Despite these colocalizations, the majority of PHF8-PS proteins remained diffused within the cytoplasm. Together our results revealed that, contrary to the parental PHF8, PHF8-PS is a protein localized in the cytoplasm of the cell.

### 3.5. PHF8-PS Specifically Interacts with Cytosolic Proteins

Since PHF8-PS did not exhibit the same subcellular localization than PHF8, we hypothesized that PHF8-PS might interact with cytoplasmic proteins and hence could potentially acquire specific cytoplasmic-related functions through its novel associated partners. To investigate, PHF8 and PHF8-PS complexes were purified in HEK293 cells and subsequently analyzed by MS. Mock experimental controls corresponding to HEK293 cells without any overexpression of PHF8 and PHF8-PS were also analyzed by MS. PHF8 and PHF8-PS protein interactions were scored using the SAINT algorithm, and the fold change over the experimental controls was evaluated using the mean of replicates and a stringent background estimation. We identified 68 proteins that associated with both PHF8 and PHF8-PS, 71 proteins that specifically interacted with PHF8, and 45 proteins that exclusively associated with PHF8-PS (Figure 5A,B; Appendix A). We next performed a gene ontology (GO) analysis to identify in which specific pathways PHF8- and PHF8-PS-associated proteins were specifically and distinctly involved. Using ShinyGO, a tool for in-depth analysis of gene sets [46], we showed that PHF8-interacting proteins were mainly involved in meiotic chromosome condensation (Figure 5C), such as SMC1A, SMC2, SMC6, Rad50, and NCAPD3 proteins (Figure 5A; Appendix A). The functions of protein specifically interacting with PHF8 correlated with its nuclear localization (Figure 4C). We also discovered that PHF8-PS-interacting proteins were enriched in prefoldin subunits that form a co-chaperone complex responsible for delivering unfolded proteins to the cytosolic chaperonin complex (Figure 5D) [47,48]. PHF8-PS specifically interacted with four of six different prefoldin subunits (PFDN2, PFDN3/VBP1, PFDN5, PFDN6) (Figure 5A; Appendix A). Moreover, gene set enrichment analyses with the cellular component ontology revealed that PHF8-PS-specific associated proteins were mainly proteins from the mitochondria membrane (Figure 5E), such as Rhot2, Ndufa10, Ambra1, Aldh18a1, Maip1, and Ndufs3 proteins (Figure 5A; Appendix A). These proteins specifically interacting with PHF8-PS supported its colocalization with Tom20 at the mitochondria (Figure 4C). Together our results showed that PHF8-PS specifically interacted with cytosolic partners, some of them being mitochondrial proteins and others prefoldin subunits helping the chaperonin complex to fold cytosolic proteins.

## 4. Discussion

Most of pseudogenes are generated by reverse-transcription and integration of an RNA transcript into the genome. Hence, these retrotransposed pseudogenes are usually intronless. We reported the presence of *Phf8-ps*, a *Phf8* retransposed pseudogene, in the mouse genome. We showed that *Phf8-ps* is indeed transcribed and translated into a protein highly homolog to the PHF8 parental protein. Interestingly, this pseudogene is not present in the human genome, raising the question of the importance of PHF8-PS. In humans, *PHF8* mutations targeting the JmjC domain have been associated to intellectual disability and craniofacial deformities [31,37,38,39]. *PHF8* is evolutionarily conserved and, for instance, injection of a *zPHF8* morpholino in the zebrafish caused delay in brain and craniofacial development, a phenotype that is similar to patients with *PHF8* mutations [31]. Surprisingly, *Phf8* knockout (KO) in mice does not lead to clinical symptoms associated with PHF8 patients such as craniofacial deformities. Loss of Phf8 only confers resistance to depression-like and anxiety-like behaviors in mice [49] and causes deficient learning and memory by epigenetic disruption of mTOR signaling [50]. The presence and potential expression of *Phf8* pseudogene in these KO mice might somehow compensate the loss of Phf8 and could explain why developmental brain and craniofacial abnormalities are not observed in these mice.

We investigated *Phf8-ps* expression in different tissues (i.e., brain, heart, spleen, lung, liver, and testis) and showed that *Phf8-ps* was expressed at RNA and protein levels mainly in testis (Figure 2). However, we only analyzed *Phf8-ps* expression in three brain areas (i.e., cortex, hindbrain, and olfactory bulbs) and we do not exclude that *Phf8-ps* could also be expressed in other brain regions. Indeed, in silico analyses indicated that brain and testis shared the highest similarity of gene expression patterns [51]. Brain and testis exhibit cellular and molecular similarities, which could be responsible for the expression of a large number of common proteins [52]. The similarity of gene expression between brain and testis is widely present in mammals, including rodents [51]. We confirmed that *Phf8-ps* was translated in vivo by identifying a unique peptide of PHF8-PS protein in the mouse germ cell proteome, more specifically in the spermatid (post-meiotic) step of spermatogenesis (Figure 2E). Previous studies have shown that histone demethylases are essential for spermatogenesis. For instance, the JHDM2A/JMJD1A histone demethylase is critical for spermiogenesis by regulating the expression of specific genes required for the final stages of sperm chromatin condensation and maturation [53]. Histone demethylases and other epigenetic players, such as writers, readers, or erasers, are important during spermatogenesis to promote histone substitution by protamines, and hence facilitate chromatin packaging, DNA silencing, and imprinting within the sperm cell [54,55]. Indeed, PHD finger protein 7 (PHF7), a member of the PHD reader family, is also a critical factor for histone-to-protamine exchange and chromatin condensation in early condensing spermatids [56,57]. To address PHF8-PS functionality, we investigated its subcellular localization and its associated proteins. We showed that PHF8-PS was mainly in the cytoplasm probably because for the absence of NLS signal (Figure 4) and that PHF8-PS interacted with specific cytosolic proteins (Figure 5A). For instance, we found that PHF8-PS was associated with four of six different prefoldin subunits (i.e., PFDN2, PFDN3/VBP1, PFDN5, and PFDN6). These prefoldin subunits form a hexameric co-chaperone complex responsible for delivering unfolded proteins to the cytosolic chaperonin complex CCT (chaperonin-containing t-complex polypeptide 1) that assists the folding of actin, tubulin, and other cytosolic proteins [47,48]. Interestingly, PFDN5-mutant mice exhibit not only neurodegeneration in the cerebellum [58], but also testicular atrophy and defective spermatogenesis including a sloughing of spermatocytes and spermatids [59]. It has been shown that diverse molecular chaperone families orchestrate the differentiation of male germ cells during spermatogenesis and that the CCT complex is essential for the cytodifferentiation of spermatids [60]. This complex acts as the folding machinery for 5 to 10% of newly synthesized cytosolic proteins [61]. Therefore, by interacting with prefoldin subunits, PHF8-PS might guide newly synthesized proteins that have not completed their folding to interact with the cytosolic chaperonin complex CCT.

We also showed that PHF8-PS colocalized with mitochondria (Figure 4C) and interacted with proteins from the mitochondrial membrane such as Rhot2, Ndufa10, Ambra1, Aldh18a1, Maip1, and Ndufs3 (Figure 5A). Spermatogenesis requires highly coordinated mitochondrial activities [62], and male infertility is associated with a loss of mitochondrial proteins [63]. Interestingly, PHF8 has been shown to regulate the mitochondrial unfolded protein response (UPR) [64]. However, it is not clear how mitochondrial stress is sensed by PHF8. One hypothesis is that PHF8 is the target of the mitochondrial-derived stress signal and modulates a transcriptional response in the nucleus. Another hypothesis could be that PHF8-PS, which is colocalized with mitochondria, plays a direct role in sensing misfolded proteins in mitochondria, potentially through its interaction with prefoldin subunits and other mitochondrial proteins. Both hypotheses are not necessarily mutually exclusive. We confirmed that PHF8 was nuclear and associated with proteins implicated in the condensation of meiotic chromosomes. It is reasonable to imagine that the nuclear PHF8 and the cytoplasmic PHF8-PS could communicate to regulate stress response pathways (i.e., mitochondrial UPR) during spermatogenesis [65]. This interplay could explain why the expression of both *Phf8* and *Phf8-ps* transcripts is so high in testis compared to other organs (Figure 2A,B).

For future research perspectives, it will be beneficial to further decipher PHF8-PS molecular underpinnings. Sequence alignment indicated that the *Phf8-ps* probably derived from the retrotransposition of *Phf8* transcript 2 because of the presence of exon 13 (Figure 1A). However, *Phf8* transcript 1 appeared to be the most expressed transcript in many tissues (Figure 2A). Interestingly, it was initially proposed that transcripts giving rise to retrotransposed/processed pseudogenes were produced in germ line cells [66]. To verify this hypothesis, it will be pertinent to determine whether the exon 13 is present in the *Phf8* transcript expressed in germ line cells. Moreover, establishing the exact contribution of this alternative exon to PHF8-PS interaction with cytosolic proteins could also be important to clarify whether this domain is critical for the recruitment of specific cytosolic proteins. Moreover, we demonstrated that PHF8-PS did not have the intrinsic capability to demethylate H3K9me2 in vitro (Figure 3). It will be interesting to address whether PHF8-PS can demethylate non-histone substrates like other demethylases [67]. Finally, it will be particularly fundamental to investigate why this *Phf8* pseudogene is solely present in the mouse genome and has not been conserved through evolution in the human genome.

## Figures and Tables

**Figure 1 genes-14-00172-f001:**
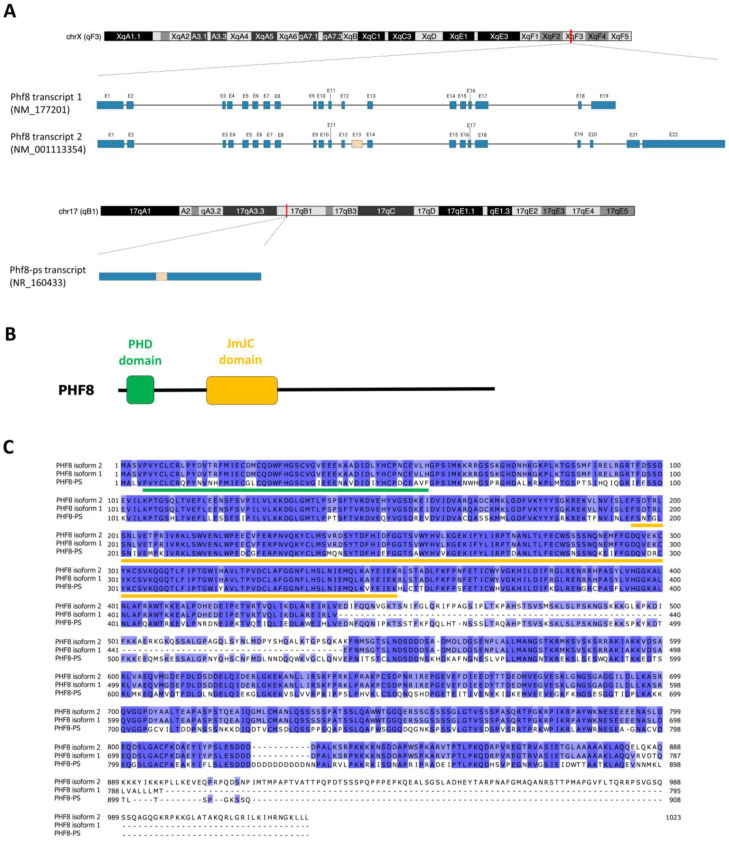
Characteristics of the mouse *Phf8* and *Phf8-ps* genes. (**A**) Schematic depicting the chromosomic location of *Phf8* and *Phf8-ps* genes in the mouse genome, and the respective transcripts produced by these genes. Exons are represented in blue boxes (excepted alternative exon 13 in beige) and introns by a line (image created with Biorender.com). (**B**) Schematic diagram of PHF8 protein structure with the functional PHD and JmjC domains. (**C**) Alignment of protein sequences of PHF8-PS and PHF8 isoforms. Sequences were obtained from the NCBI Ref seq database. Sequence alignments were performed using Jalview (version 2.11.2.3) and the Clustal Omega algorithm with default parameters. The conserved PHD and JmjC domains are respectively underlined in green and orange.

**Figure 2 genes-14-00172-f002:**
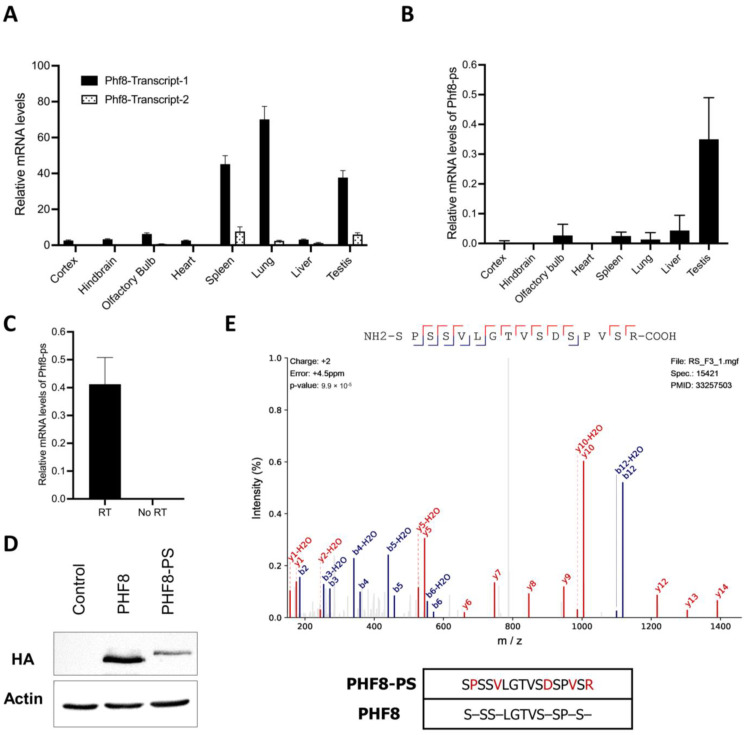
Phf8-ps mRNA and protein expression. (**A**,**B**) Relative mRNA levels of (**A**) Phf8 transcripts 1 and 2 and (**B**) Phf8-ps transcript were measured in different mouse organs or tissues by quantitative PCR and normalized to that of GAPDH (n=3 per organ/tissue). (**C**) Relative mRNA levels of Phf8-ps transcript were measured by quantitative PCR using on cDNA from testis samples prepared with or without reverse transcriptase. Levels were normalized to mRNA levels of GAPDH (n = 3 per condition). (**D**) NIH3T3 cells were transfected to transiently overexpress HA/FLAG-tagged PHF8 and PHF8-PS. Protein levels of PHF8 and PHF8-PS were analyzed by Western blotting as indicated. (**E**) Peptide spectrum of a unique PHF8-PS peptide (SPSSVLGTVSDSPVSR). This unique peptide was found in the dataset of mouse germ cell proteome [40]. The sequence homology between this PHF8-PS peptide and PHF8 protein is represented at the bottom panel.

**Figure 3 genes-14-00172-f003:**
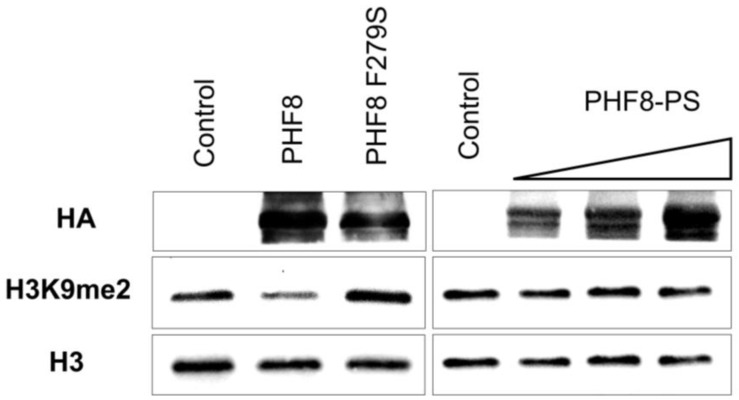
Characterization of PHF8-PS demethylase activity. Histone demethylase assays on bulk histones with PHF8 and PHF8-PS proteins purified from HEK293 cells overexpressing Flag/HA-tagged PHF8, PHF8 catalytic mutant (PHF8 F279S), and PHF8-PS. PHF8-PS demethylase assays were performed with an increasing number of proteins. Proteins were purified on FLAG tag and analyzed by Western blotting as indicated.

**Figure 4 genes-14-00172-f004:**
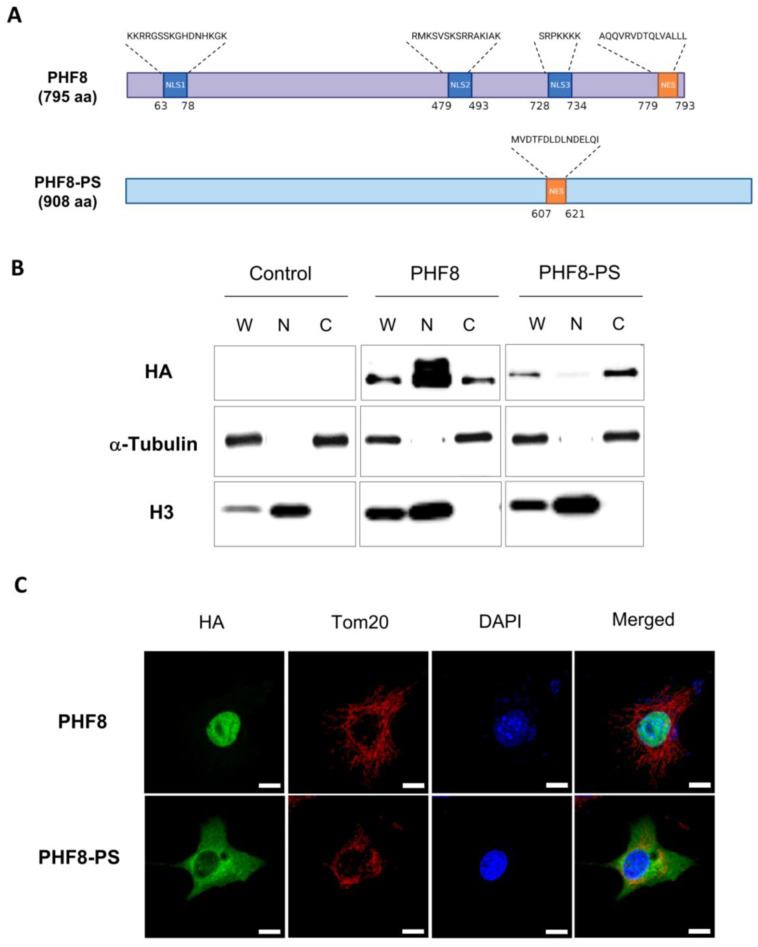
Subcellular localization of PHF8-PS. (**A**) Schematic representation of nuclear localization signals (NLS) and nuclear export signals (NES) in PHF8 and PHF8-PS proteins (image created with Biorender.com). (**B**) Cell fractionation of NIH3T3 cells transiently expressing HA/FLAG-tagged PHF8 and PHF8-PS proteins. Whole cell lysate (W), nuclear (N), and cytoplasmic (C) fractions were analyzed by Western blot as indicated. (**C**) Co-immunostaining images of NIH3T3 cells transiently expressing HA/FLAG-tagged PHF8 and PHF8-PS showing the subcellular localization of these proteins (green) and that of the mitochondrial marker Tom20 (red). Cell nuclei were stained DAPI (blue). Scale bar: 10 μm.

**Figure 5 genes-14-00172-f005:**
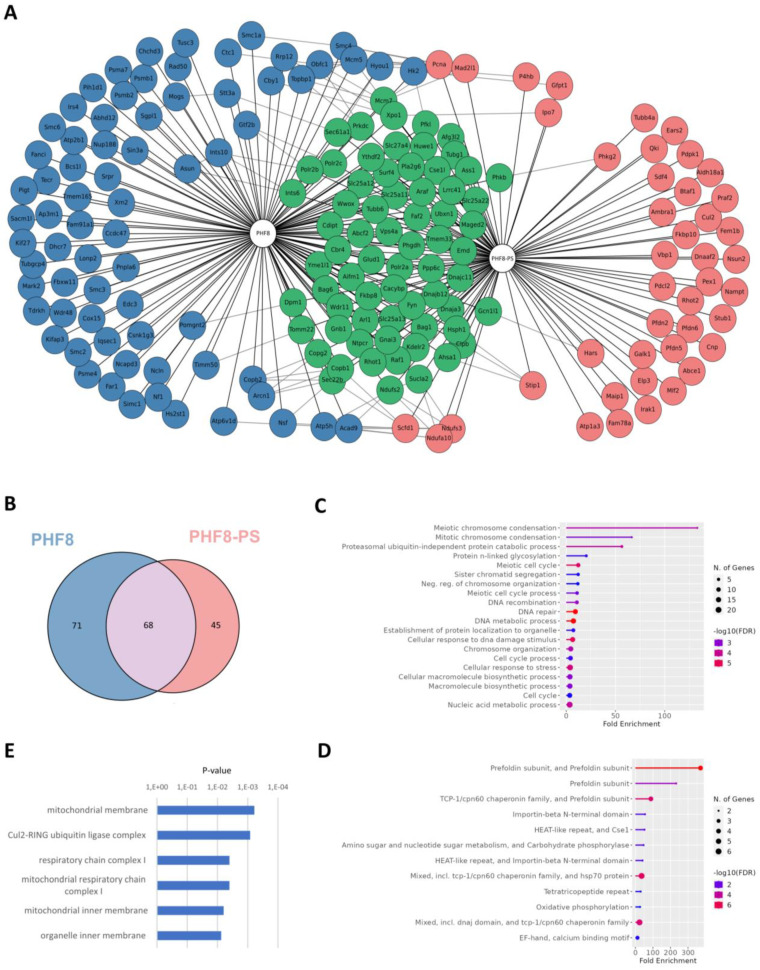
Identification of PHF8-PS-interacting proteins. (**A**) Schematic representation of PHF8- and PHF8-PS-interacting network. PHF8 and PHF8-PS complexes were purified in HEK293 cells transiently expressing these proteins. Protein interactions retrieved from the STRING database were indicated. (**B**) Venn diagram showing the overlap between PHF8- and PHF8-PS-interacting proteins. (**C,D**) Gene Ontology (GO) analysis of (**C**) PHF8-specific and (**D**) PHF8-PS-specific interacting proteins using the ShinyGO tool (version 0.76.1). (**E**) Gene Set enrichment analysis (GSEA) of PHF8-PS-specific interacting proteins with the cellular component ontology. Cellular components are sorted by *p*-value ranking.

**Table 1 genes-14-00172-t001:** List of antibodies used for immunoblotting.

Antibody	Company	Reference	Dilution
Anti-HA	Roche (Basel, Switzerland)	11867423001	1:500
Anti-tubulin	Santa Cruz Biotechnology, Inc. (Dallas, TX, USA)	sc-23948	1:1000
Anti-H3K9me2	MilliporeSigma (Burlington, MA, USA)	07-441	1:5000
Anti-H3	Abcam (Cambridge, UK)	Ab1791	1:5000
Anti-mouse HRP	Biorad (Hercules, CA, USA)	170-6516	1:5000
Anti-rabbit HRP	Biorad (Hercules, CA, USA)	170-6515	1:5000

## Data Availability

Raw MS files will be shared on reasonable request to the corresponding author.

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
