# Peer review of "Functional Characterization of a Phf8 Processed Pseudogene in the Mouse Genome"

_genes, 2023, doi:10.3390/genes14010172_

Round 1
Reviewer 1 Report
The paper is very well written. The experiments are solid and well-presented. I would like to suggest to the authors to include one paragraph in the discussion focusing on the know role and functions of PHF8 transcript 2. This is because PHF8-ps sequence is more similar to the PHF8 transcript 2. Otherwise, interesting and good data are presented in the manuscript.
Author Response
We thank the reviewer for her/his positive comments. As suggested, we have further developed the discussion on the potential functions of Phf8 transcript 2 in germ line cells.
Reviewer 2 Report
St-Germain and colleagues investigated potential existence of a functional protein encoded by Phf8-ps, which is currently annotated as a pseudogene. The authors analyzed a previously published dataset and discovered unique peptides of Phf8-ps in mouse spermatids. They further demonstrated that unlike PHF8, Phf8-ps does not harbor a nuclear localization signal or show demethylase activity in vitro. Interestingly, by co-IP/MS, the authors found that Phf8-ps specifically interacts with prefoldin subunits and mitochondrial membrane proteins. This work provides insight into the possible molecular function of Phf8-ps and potentially attracts wide attention from cell and developmental biologists. The manuscript could be further improved by addressing the issues as listed below:
1. Line 17: Since the authors are referring to the murine ortholog, PHF8 and PHF8-ps should be changed to Phf8 and Phf8-ps. Please correct the gene symbols in the manuscript.
2. Line 18: Phf8-ps is not “specifically” transcribed in testis. It shows minor expression in other tissues.
3. Line 21: Change “Phf8 protein” to “PHF8 protein”. Please make sure protein symbols are in uppercase throughout the text (e.g., Lines 79, 82, 464 – 465).
4. Given the prominent expression of PHF8 in testis, it would be beneficial to introduce the important roles that histone demethylases play in spermatogenesis (e.g., PMID: 17943087).
5. In the left panel of Figure 3, PHF8 carrying a F279S mutation shows a reduced level of expression compared to wildtype PHF8. Thus, it would be difficult to conclude whether the reduced demethylation was caused by the decreased protein level or impaired demethylase activity.
6. In the right panel of Figure 3, the protein bands in H3K9me2 and H3 immunoblots look very similar. In Figure 4B, small dots with a straight vertical edge are observed in the middle lanes of the three alpha-tubulin blots. The authors should provide unprocessed, uncropped raw Western blot images in the supplementary materials.
7. Phf8-ps does not exhibit demethylase activity probably due to its predominant localization in the cytoplasm. If you replace its NES with an NLS, will this protein show demethylase activity?
8. In Figure 4C, Phf8-ps seems to be strongly expressed in the ER (rather than mitochondria). Please test the possible co-localization between Phf8-ps and ER markers.
Author Response
We thank the reviewer for her/his constructive criticisms and helpful suggestions. We have addressed all the comments. The detailed point-by-point responses are provided below.
- Line 17: Since the authors are referring to the murine ortholog, PHF8 and PHF8-psshould be changed to Phf8 and Phf8-ps. Please correct the gene symbols in the manuscript.
As requested by the reviewer, we modified the gene symbols throughout the manuscript.
- Line 18: Phf8-ps is not “specifically” transcribed in testis. It shows minor expression in other tissues.
The reviewer is correct; we changed “specifically” by “mainly” in the revised manuscript.
- Line 21: Change “Phf8 protein” to “PHF8 protein”. Please make sure protein symbols are in uppercase throughout the text (e.g., Lines 79, 82, 464 – 465).
As requested by the reviewer, we changed the protein symbols in the manuscript and all the figures.
- Given the prominent expression of PHF8 in testis, it would be beneficial to introduce the important roles that histone demethylases play in spermatogenesis (e.g., PMID: 17943087).
Following the reviewer’s suggestion, we included a paragraph in the discussion section on the involvement of histone demethylases and other epigenetic regulators in spermatogenesis.
- In the left panel of Figure 3, PHF8 carrying a F279S mutation shows a reduced level of expression compared to wildtype PHF8. Thus, it would be difficult to conclude whether the reduced demethylation was caused by the decreased protein level or impaired demethylase activity.
The critical amino acids for PHF8 demethylase activity have been extensively investigated. The F279S mutation was identified in PHF8 from a family of XLMR patients (PMID: 17661819) and is known to modify the conserved hydrophobic pocket, leading to this PHF8 mutant to be catalytically inactive (PMID: 19843542). In vitro enzymatic assays with WT and F279S mutant PHF8 have already been performed (PMID: 20622853) and are used here as a control to confirm the enzymatic activity of the purified PHF8.
- In the right panel of Figure 3, the protein bands in H3K9me2 and H3 immunoblots look very similar. In Figure 4B, small dots with a straight vertical edge are observed in the middle lanes of the three alpha-tubulin blots. The authors should provide unprocessed, uncropped raw Western blot images in the supplementary materials.
In Figure 3, H3K9me2 and H3 immunoblots are similar because these two antibodies were used to probe the same membrane. The membrane was first probed and revealed for H3K9me2, and then both primary and secondary antibodies were removed from the membrane with a stripping solution before reprobing with H3 antibody. We usually apply this methodology for important experiments to ensure that observed results do not result from a misloading between two different immunoblots. We added this methodology in the material and Methods section.
In Figure 4B, the small dots come from the ECL used to reveal the immunoblot. In the case of strong antibody signals (e.g. actin, tubulin antibodies) and to avoid a quick saturation of the signal, we do a homemade ECL (luminol, p-coumaric acid, H2O2), which is less strong than the commercial ones. However, this homemade ECL sometime results in small dots on the immunoblots.
As requested by the reviewer, we added all the raw immunoblot images in the supplement material (Supplemental Figure 2).
- Phf8-ps does not exhibit demethylase activity probably due to its predominant localization in the cytoplasm. If you replace its NES with an NLS, will this protein show demethylase activity?
Histone demethylases remove methyl groups on histones but also on non-histone substrates. Their enzymatic activity does not depend necessarily on the subcellular localization. In vitro demethylase assays are usually performed to determine the intrinsic activity of the enzyme to demethylate a substrate whether it is in the nucleus or the cytoplasm. The results from Figure 3 indicate that PHF8-PS does not have the intrinsic property to demethylate histones.
- In Figure 4C, Phf8-ps seems to be strongly expressed in the ER (rather than mitochondria). Please test the possible co-localization between Phf8-ps and ER markers.
As suggested by the reviewer, we performed immunofluorescence against endoplasmic reticulum (ER) markers (i.e., P4HB, Calnexin) (Supplemental Figure 1). A small fraction of PHF8-PS proteins was colocalized with these markers at the endoplasmic reticulum while the majority of PHF8-PS remained diffused within the cytoplasm.
Round 2
Reviewer 2 Report
The authors have fully addressed my concerns. I support publication of this manuscript in its present form.